# Challenging the Counterintuitive: Revisiting Simple Likelihood Tests with Normalizing Flows for Tabular Data Anomaly Detection

## Abstract

In this study, we propose a novel approach to anomaly detection in the tabular domain using normalizing flows, leveraging a simple likelihood test to achieve state-of-the-art performance in unsupervised learning. Although simple likelihood tests have been shown to fail in anomaly detection for image data, we redefine the counterintuitive phenomenon and demonstrate, both theoretically and empirically, why this method succeeds in the tabular domain. Our approach outperforms traditional anomaly detection methods by offering more consistent results. Furthermore, we question the practice of fine-tuning parameters for each dataset individually, ensuring fair and unbiased comparisons by adopting uniform hyperparameters across all datasets. Through extensive experimentation, we validate the robustness and scalability of our method, highlighting its practical effectiveness in real-world settings.

## 1 Introduction

Generative models with artificial neural networks are models that learn the distribution of data and are actively researched and utilized in diverse applications such as the generation of industrial data, medical data, etc. An example of such a model is the variational autoencoder (Kingma, 2013), which adopts Evidence Lower BOund(ELBO) as an objective function and maximizes it to indirectly maximize the log-likelihood $\log p(x)$ for input data $x \in \mathbb{R}^d$, normalizing flow (Dinh et al., 2014); in addition it utilizes, an invertible function $f$ to transform the distribution of complex input data into a simple distribution such as the Standard Gaussian Distribution, as well as generative adversarial networks (Goodfellow et al., 2020), where a network comprising a discriminator and two generators is trained to minimize/maximize the minimax loss, respectively. Models that can obtain the true likelihood of the input data, such as normalizing flow, can be utilized in the field of anomaly detection or out-of-distribution detection (OOD Detection) through likelihood[1]. The simplest approach of anomaly detection using normalizing flow is to assume that normal data $x \in \mathbb{R}^d$ follows the distribution $P$ of normal data, and anomalous data $x' \in \mathbb{R}^d$ follows a distribution $P \neq Q$, and to determine that a given data $x_{test} \in \mathbb{R}^d$ is an anomaly if its likelihood $\phi_P(x)$ is lower than a predefined threshold $\alpha$ when tested.

This methodology is based on the intuition that anomalous data are less likely to be observed in the distribution of normal data. However, the image domain illustrates that in-distribution data utilized as training data in models that can obtain the likelihood of the input data indirectly or directly, such as normalizing flow, exhibit similar or even lower likelihoods than out-of-distribution data. For example, when CIFAR10 (Alex, 2009) is used as training data (In-distribution) and SVHN (Netzer et al., 2011) is used as the test data (Out-of-distribution) of a model that can obtain the likelihood of input data, SVHN has a higher likelihood than CIFAR10 (Nalisnick et al., 2018). Kirichenko et al. (2020) analyzes why this counterintuitive phenomenon occurs in anomaly detection for normalizing flow and suggests an approach to mitigate this phenomenon. Serrà et al. (2019), and Osada et al. (2024) experimentally and theoretically demonstrated that normalizing flow tends to assign low likelihood to complex images, i.e., images with high bit rates and compression algorithms like PNG,

---

[1]Although the two tasks slightly differ, we consider OOD detection and anomaly detection to be the same task, and we will utilize the term anomaly detection. Task definitions are presented in Appendix A.

and thereby mitigating this phenomenon by performing anomaly detection with a scoring function comprising the likelihood of the input image and the complexity of the image. In conclusion, it can be inferred that if anomaly detection is performed using only the likelihood of the input data, detection may fail in certain cases (e.g. Serrà et al. (2019); Ren et al. (2019); Nalisnick et al. (2019); Kamkari et al. (2024); Osada et al. (2024). Refer to Section 2.2 for more details.

However, the following question arise: does this phenomenon also occur in tabular data anomaly detection? Kirichenko et al. (2020) demonstrates that although the likelihood of in-distribution/OOD data overlaps for the normalizing flow in the tabular data anomaly detection, it is limited by the fact that only two datasets are shown by setting each as in-distribution data/OOD data. In addition, there is no comparison with other comparison models; hence, it is not known whether the counter-intuitive likelihood phenomenon in the tabular domain occurs. Accordingly, we present a general definition of the counterintuitive phenomenon of likelihood-based tests for models that provide an exact likelihood to apply it to all domains, conduct an extended experiment to verify whether the simple likelihood test method, which has been described as exhibiting several limitations in the image anomaly detection, is valid in the tabular data anomaly detection. Consequently, we empirically demonstrate that almost all datasets do not exhibit counterintuitive phenomena in the tabular domain, and even outperform other models in simple likelihood tests. Furthermore, we theoretically demonstrates the case to support our claim.

In a real unsupervised setting, we train only normal datasets, limiting the optimal hyperparameter selection. However, previous papers that performed tabular anomaly detection under unsupervised settings have selected different hyperparameters for each dataset when evaluating the performance of each dataset (except for the hyperparameter-free model, e.g. Li et al. (2020)). Therefore, it is difficult to suggest that the experiments performed in such a setting in previous studies and showing the performance by selecting the optimal hyperparameter for each dataset truly perform experiments under an unsupervised setting. The basis for this claim is that most unsupervised anomaly detection methods' dataset split protocol follow Zong et al. (2018), which does not set up a validation dataset for hyperparameter selection. Therefore, inspired by the biased hyperparameter selection of current experiments in tabular anomaly detection, we conduct a fair experiment that selects hyperparameters from a predefined hyperparameter searching space and evaluates their performance by applying them to all datasets. In conclusion, the contribution of our study can be described as threefold.

- First, we provide a general definition of the counterintuitive phenomenon in likelihood-based tests and empirically show that simple likelihood testing using normalizing flows in the tabular domain rarely leads to this phenomenon. We also provide a theoretical analysis to support this.

- Second, we point out biased and impractical hyperparameter selection in tabular anomaly detection tasks and conduct fair hyperparameter selection and performance evaluation that is suitable for unsupervised settings.

- Third, compared to previous studies, we verify our results by conducting experiments with an extended dataset, avoiding biased dataset selection (refer to Section 4.1). Furthermore, our approach demonstrates state-of-the-art performance when applied to tabular anomaly detection tasks using 47 tabular benchmark datasets. This supports the superiority of the simple likelihood test using normalizing flows in the tabular dataset.

## 2 RELATED WORK

### 2.1 NORMALIZING FLOW

Normalizing flow is one of the generative models that converts input data $\mathbf{x} \in \mathbb{R}^d$, which follows an unknown distribution called $p_\mathbf{x}$, into $\mathbf{z} \in \mathbb{R}^d$; in addition, it follows a simple distribution $p_\mathbf{z}$ that typically selected standard Gaussian $\mathcal{N}(0, I_d)$ (Dinh et al., 2016), using an invertible function $f : \mathbb{R}^d \to \mathbb{R}^d$ that consists of complex functions such as neural networks (Dinh et al., 2014), such that $p_\mathbf{x}$ can be expressed as a formula expressed in terms of $p_\mathbf{z}$. At this point, $p_\mathbf{x}$ is expressed as the determinant of Jacobian of $\mathbf{x}$ and $\mathbf{z}$ and $p_\mathbf{z}$ by the change-of-variable rule, and is expressed as Equation 1.

$$\log p_\mathbf{x}(\mathbf{x}) = \log p_\mathbf{z}(\mathbf{z}) + \log |\boldsymbol{J}|, \boldsymbol{J} = \det \frac{\partial \mathbf{z}}{\partial \mathbf{x}} \tag{1}$$

In general, it learns in the direction of maximizing the likelihood $\log p_{\mathbf{x}}(\mathbf{x})$ of the learning input data, and approximates the distribution of the input data (Caterini & Loaiza-Ganem, 2022). The normalizing flow model can be categorized into a model whose volume has a constant-volume term that is invariant to the input data (e.g. Dinh et al. (2014)) and variant to the input data (e.g. Rezende & Mohamed (2015); Dinh et al. (2016); Kingma & Dhariwal (2018); Behrmann et al. (2019); Chen et al. (2019); Durkan et al. (2019)). When sampling new data, sampling is performed by extracting it from the pre-defined $p_{\mathbf{z}}$ and inputting it as the input of $f^{-1}$. The normalizing flow has the advantage of being able to obtain the actual likelihood of the input data, unlike models such as variational autoencoder and generative adversarial network, it has the advantage of not requiring the additional likelihood approximate inference techniques (Nalisnick et al., 2018). However, normalizing flow has two constraints: 1) the computational amount of Jacobian must not become too large, and 2) the inverse of $f$ must exist. Therefore, the following methodologies were utilized to ensure the ease of Jacobian calculation and the existence of the inverse $f^{-1}$: 1) introducing a special architecture such as a coupling layer (Dinh et al., 2014; 2016; Kingma & Dhariwal, 2018), 2) using a function of a special form (Rezende & Mohamed, 2015), 3) using the approximation method of power series and the Lipschitz constraint (Behrmann et al., 2019; Chen et al., 2019), etc. In our study, we demonstrated that NICE (Dinh et al., 2014), a relatively simple model, outperforms other comparison models when performing a simple likelihood test. We refer to this methodology using normalizing flow as NF-SLT (Normalizing Flow with Simple Likelihood Test).

## 2.2 Counterintuitive Phenomenon of Likelihood

Nalisnick et al. (2018) reported that a counterintuitive phenomenon regarding likelihood assignment occurs in models that can obtain exact likelihood, such as normalizing flow, in the image domain. This study lays the foundation for identifying the cause of this phenomenon or suggesting solutions. Kirichenko et al. (2020); Schirrmeister et al. (2020) improved performance by changing the structure of the existing flow model. In Addition, the latter proposed a methodology to improve performance by reflecting the hierarchical structure of data. Serrà et al. (2019) quantified complexity through a general compression algorithm such as PNG, based on experimental results, demonstrating that simple images exhibit higher likelihood, and presented an anomaly score combining the likelihood and complexity terms. Kamkari et al. (2024) used Local Intrinsic Dimension (LID) to measure an image's simplicity and proposed a dual thresholding method for LID and likelihood to improve performance. Morningstar et al. (2021), Osada et al. (2024), and Ahmadian et al. (2021) are improved the disadvantages of using only a single likelihood by estimating the density of a vector that combines the likelihood and several statistics (e.g. complexity, determinant of Jacobian). Nalisnick et al. (2019) demonstrated the perspective that detection may fail because in-distribution data are located in the typicality set (Cover, 1999) and OOD data is in the high density set. Zhang et al. (2021) presented the view that the counterintuitive phenomenon occurs due to misestimation of the model. Le Lan & Dinh (2021) demonstrated that even with a perfect model, simple likelihood-based methods can fail due to variants in the representation. Ren et al. (2019) improved performance by using the likelihood ratio between the background and semantic models. and Caterini & Loaiza-Ganem (2022) explained the cause of this phenomenon from an entropic perspective and why the likelihood ratio model works well. We define a general counterintuitive phenomenon that can be applied to all domains, which demonstrates that a simple likelihood-based method that adopts a flow model in the tabular domain outperforms other existing methods, showing "**counter of counterintuitive**" phenomenon in which the counterintuitive phenomenon do not occur or rarely occur unlike image domain through extended experiments and theoretical analysis.

## 3 Definition of Counterintuitive Phenomenon

Anomaly detection methods based on likelihood tests have revealed surprising results in certain cases, commonly referred to as the counterintuitive phenomenon. This occurs when out-of-distribution (OOD) data are assigned higher likelihoods than in-distribution data, contradicting the expected behavior of likelihood-based models. Nalisnick et al. (2018) first observed this issue when models trained on CIFAR10 images as in-distribution data assigned higher likelihoods to OOD data such as SVHN. These findings challenge the assumption that normal data should have higher likelihoods than anomalous data. While the counterintuitive phenomenon has been extensively studied in image datasets, its prevalence in tabular data remains uncertain. Earlier research (Kirichenko et al.,

2020) noted instances where in-distribution and OOD data had overlapping likelihoods in tabular datasets, but these findings were limited to only a few datasets and lacked comprehensive comparisons with other anomaly detection models. To address this gap, we propose a generalized definition of the counterintuitive phenomenon that applies to diverse tabular datasets.

To formalize this phenomenon, we begin by establishing two core assumptions:

**Assumption 1.** *If a counterintuitive phenomenon occurs, most comparison models should outperform the generative model on an anomaly detection task.*

**Assumption 2.** *Even if the above condition is satisfied, the performance gap between the generative model and comparison models must be significant to qualify as a counterintuitive phenomenon. If the gap is small, it cannot be considered counterintuitive.*

We now formalize this phenomenon using these assumptions.

**Definition 1** (Occurrence of Counterintuitive Phenomenon). *Let $x \sim P$, where $P_{\theta_0}$ provides an approximately exact likelihood estimate of the input $x$, and let $P_{\theta_k}$ represent $k$ comparison models that do not provide likelihood estimates. Assume all models are well-trained.*

*Let $\varphi_{P_{\theta_0}}(x)$ represent the likelihood estimate from the generative model $P_{\theta_0}$, and let $\varphi_{P_{\theta_k}}(x)$ represent the test statistic (e.g., anomaly score) from the $k$-th comparison model. Define $\alpha_k \in \mathbb{R}$, $\beta, \gamma \in (0, 1]$ as predefined thresholds. Let*

$$R = \left\{ i \in [k] \mid \Pr(\varphi_{P_{\theta_0}}(x) > \varphi_{P_{\theta_0}}(y)) < \Pr(\varphi_{P_{\theta_i}}(x) > \varphi_{P_{\theta_i}}(y)) \right\} \tag{2}$$

*The hypothesis test for anomaly detection can be formalized as:*

$$\begin{aligned} H_0 &: x \sim P \\ H_1 &: x \sim Q \quad \text{such that } P \neq Q, \end{aligned} \tag{3}$$

*where $Q$ is the distribution of anomalous data. If $\varphi_{P_k}(x) < \alpha_k$, reject $H_0$.*

*We define that a counterintuitive phenomenon occurs if the following two conditions are satisfied:*

*1. The majority of comparison models outperform the generative model:*

$$\frac{1}{k} \sum_{i=1}^{k} 1 \left\{ \Pr(\varphi_{P_{\theta_0}}(x) > \varphi_{P_{\theta_0}}(y)) < \Pr(\varphi_{P_{\theta_i}}(x) > \varphi_{P_{\theta_i}}(y)) \right\} > \beta, \quad y \sim Q. \tag{4}$$

*2. The minimum performance gap between the generative model and the outperforming comparison models is significant:*

$$\min_{i \in R} \left( \Pr(\varphi_{P_{\theta_i}}(x) > \varphi_{P_{\theta_i}}(y)) - \Pr(\varphi_{P_{\theta_0}}(x) > \varphi_{P_{\theta_0}}(y)) \right) > \gamma, \quad y \sim Q. \tag{5}$$

Definition 1 states that a counterintuitive phenomenon occurs when the proportion of cases in which the AUROC of $k$ comparison models is higher than that of the generative model $P_\theta$, as tested by the hypothesis in Equation 3, exceeds $\beta$, and the minimum AUROC difference between $P_\theta$ and the models that outperform $P_\theta$ is greater than $\gamma$.

Consider the CIFAR-10 (in-distribution) vs. SVHN (out-of-distribution) example. According to Morningstar et al. (2021), a simple likelihood test using the Glow model (Kingma & Dhariwal, 2018) yielded an AUROC of 6.4%. In contrast, Sun et al. (2022) achieved AUROC scores exceeding 90% with their proposed method and comparison models. Based on Definition 1, this case clearly demonstrates a counterintuitive phenomenon, as the generative model performs significantly worse than the comparison models. To explore whether this phenomenon occurs in tabular data, we conducted experiments to test if a counterintuitive phenomenon, as defined in Definition 1, appears in tabular anomaly detection datasets. Some might argue that assigning higher likelihoods to anomalies than to normal data suggests a counterintuitive phenomenon, but this view is incomplete. It focuses only on individual data points rather than overall model performance. The occurrence of a counterintuitive phenomenon must be determined by comparing the generative model's performance against other models (e.g., $k$-NN, OCSVM), not simply by an AUROC lower than 100%. Without this comparative analysis, it's difficult to conclusively identify the phenomenon.

## 4 EXPERIMENT

This section describes the experiment setting and results, including how we ensure fair hyperparameter selection.

### 4.1 EXPERIMENT SETTING

**Dataset and Preprocessing** The experiment was conducted using the data split protocol in Zong et al. (2018). To explain this protocol, 50% of normal data is used for training, and the remaining 50% of normal and abnormal data are used as test data. We used all 47 tabular datasets presented in ADBench(Han et al., 2022), except CV/NLP embedding dataset(e.g. MV-Tec Embedding). To the best of our knowledge, this is the first time we have run an experiment with all the tabular data proposed in ADBench under the data split protocol used in the experiment. This setting was motivated by Shwartz-Ziv & Armon (2022), who criticized that researchers selected datasets with selection bias to make it look like they were performing well; hence, we adopted all of the proposed benchmark datasets without selection. All models except the NeuTraLAD model utilized RobustScaler provided by the Python library Scikit-learn (Pedregosa et al., 2011) to standardize the input data. The reason for excluding NeuTraLAD is that a significant performance decrease was observed when scaling.

**Fair Hyperparameter Searching** Most of the studies in the tabular anomaly detection task follow the dataset splitting protocol of Zong et al. (2018), which is an unsupervised setting with no validation set, so choosing the optimal hyperparameter for each dataset is impossible. However, different hyperparameters were selected for each datasets to demonstrate performance, against the unsupervised setting; therefore, we conducted a fair hyperparameter search. This method first selects some hyperparameters that are considered important for each comparison model (e.g. select $k$ in $k$-NN), and then defines a discrete hyperparameter-specific searching space. Then, for each dataset, after experimenting with all combinations in the hyperparameter searching space with 10 repeated experiments, the hyperparameter combination with the highest average AUROC (Area Under Receiver Operating Characteristic Curve) for all datasets is selected as the representative hyperparameter combination to demonstrate the performance of the model. The hyperparameter search space for each model is recorded in Appendix B.

**Models** We compared the performance of 7 shallow AD models and 6 deep AD models. We implemented the shallow model using the PyOD (Zhao et al., 2019). The compared shallow models are PCA (Shyu et al., 2003), LOF (Breunig et al., 2000), IF (Liu et al., 2008), OCSVM (Schölkopf et al., 1999), $k$-NN (Ramaswamy et al., 2000), COPOD (Li et al., 2020), and ECOD (Li et al., 2022). The compared deep models are DAGMM (Zong et al., 2018), DeepSVDD (Ruff et al., 2018), GOAD (Bergman & Hoshen, 2020), NeuTraLAD (Qiu et al., 2021), ICL (Shenkar & Wolf, 2022), MCM (Yin et al., 2024), and NF-SLT with NICE (Dinh et al., 2014). Experiments on RealNVP (Dinh et al., 2016) are included in Appendix C. In addition, DO2HSC (Zhang et al., 2024) was excluded from the experiment because, after checking their implementation code, it was confirmed that the model uses the statistics of the test data when performing orthogonal projection, which was judged to be biased when compared to other models. This method first selects some hyperparameters that are considered essential for each comparison model (e.g. select $k$ in $k$-NN), and then defines a hyperparameter-specific searching space.

**Evaluation** We evaluate these AD models using AUROC and AUPRC(Area Under Precision-Recall Curve). We conducted 10 repeated experiments for each dataset, and recorded the average AUROC and AUPRC scores in Table 1. The hyperparameters that are finally selected are described in Appendix B. In addition, we recorded the absolute performance score and the relative rank of each model for relative comparison of AD models. As previously mentioned, each model searches for hyperparameters in a predefined parameter search space. When comparing the performance of each model, the selected hyperparameter with the maximum relative average AUROC rank can be considered the optimal hyperparameter for the model, to avoid selecting a hyperparameter that records very high performance only on certain datasets. However, since it is difficult to compare the performance in all hyperparameter search spaces relatively, the hyperparameter with the maximum average AUROC was recorded as the representative performance of the model. We also experimented by selecting the highest performing hyperparameter in hyperparameter searching spacefor each model for each dataset, which is an unfair hyperparameter selection method in the existing

method and recorded the results in Table 2. The AUROC and AUPRC scores for all datasets are recorded in Appendix D.

Table 1: AUROC and AUPRC performance with fair hyperparameter selection. The Avg. Rank represents the average AUROC rank of the models across all datasets. The Top2 Cum. Ratio indicates the proportion of datasets where each model ranked within the top two for AUROC, while the Fail Ratio shows the proportion of datasets where a model's AUROC rank was 9th or lower.

| Method | AUROC ↑ | AUPRC ↑ | Avg. Rank ↓ | Top2 Cum. Ratio ↑ | Fail Ratio ↓ |
|---|---|---|---|---|---|
| PCA | 0.7715 | 0.5209 | 7.13 | 0.15 | 0.45 |
| LOF | 0.8169 | 0.5606 | 6.09 | 0.15 | 0.23 |
| IF | 0.8014 | 0.5060 | 6.34 | 0.17 | 0.23 |
| OCSVM | 0.3438 | 0.2200 | 12.74 | 0.04 | 0.96 |
| $k$-NN | **0.8634** | **0.6439** | **3.28** | **0.43** | **0.00** |
| COPOD | 0.7471 | 0.4419 | 8.38 | 0.11 | 0.57 |
| ECOD | 0.7425 | 0.4530 | 8.74 | 0.09 | 0.68 |
| DAGMM | 0.6467 | 0.3468 | 11.04 | 0.00 | 0.87 |
| DeepSVDD | 0.7687 | 0.5388 | 7.68 | 0.04 | 0.40 |
| GOAD | 0.6086 | 0.4114 | 10.26 | 0.00 | 0.62 |
| NeuTraLAD | 0.8081 | 0.5694 | 6.09 | 0.23 | 0.30 |
| ICL | 0.8208 | 0.6170 | 5.70 | 0.23 | 0.26 |
| MCM | 0.7864 | 0.5383 | 7.38 | 0.09 | 0.36 |
| NF-SLT | **0.8575** | **0.6398** | **4.00** | **0.34** | **0.06** |

## 4.2 EXPERIMENT RESULT

Consider Definition 1; if a counter-intuitive phenomenon is also frequent in the tabular domain, it should have a high fail ratio even if it works well on a particular dataset resulting in a high top2 cum. ratio. In addition, the failed dataset should have a large minimum performance difference from the other models. However, based on the results in Table 1, we can observe that NF-SLT has a lower fail ratio than the shallow and deep models except $k$-NN and outperforms other metrics. Furthermore, on the "yeast" dataset where NF-SLT failed, the minimum performance difference between MCM and AUROC is 0.02%; hence, we cannot assume that it failed because of a counterintuitive phenomenon. Although the comparison model is hyperparameter sensitive and a fair hyperparameter search might lead one to believe that NF-SLT overstates its performance, the results in Table 2 show that NF-SLT outperforms the other models. Interestingly, we also observed that: $k$-NN outperforms all other models. However, $k$-NN has a disadvantage due to its high time complexity, especially during inference with large training datasets and high-dimensional inputs, making it computationally expensive and impractical for real-world applications. Additionally, we conducted a typicality test, which is an alternative method for addressing counterintuitive phenomena in the tabular domain, with the results detailed in Appendix E.

# 5  WHY IS THE SIMPLE LIKELIHOOD TESTS SUCCESSFUL IN TABULAR DATA?

This section explores both empirically and theoretically why simple anomaly detection using normalizing flow tend to work well for tabular data. In essence, tabular data generally have lower dimensionality compared to image data, which makes simple likelihood tests more effective in this context.

To illustrate how the performance of NF-SLT varies with dimensionality, we conducted some experiments with pairs of normal and anomaly data. The normal and anomaly distributions are set to Gaussian distributions but have different parameters $\mu$ and $\Sigma$. Randomly sample $10^4$ data points from the normal distribution and set them as the training dataset, learn the NICE model, and then randomly sample $10^4$ data points from the normal distribution and $10^4$ data points from the abnormal distribution and set them as the test dataset. A simple likelihood test was performed and the

Table 2: AUROC and AUPRC performance without fair hyperparameter selection

| Method | AUROC ↑ | AUPRC ↑ | Avg. Rank ↓ | Top2 Cum. Ratio ↑ | Fail Ratio ↓ |
|--------|---------|---------|-------------|-------------------|--------------|
| PCA | 0.7752 | 0.5240 | 7.55 | 0.11 | 0.49 |
| LOF | 0.8447 | 0.5979 | 6.13 | 0.21 | 0.34 |
| IF | 0.8036 | 0.5099 | 6.83 | 0.17 | 0.34 |
| OCSVM | 0.3438 | 0.2200 | 12.91 | 0.04 | 0.96 |
| $k$-NN | **0.8732** | **0.6621** | **3.51** | **0.47** | **0.06** |
| COPOD | 0.7471 | 0.4419 | 9.02 | 0.04 | 0.64 |
| ECOD | 0.7425 | 0.4530 | 9.40 | 0.02 | 0.70 |
| DAGMM | 0.6468 | 0.3473 | 11.70 | 0.00 | 0.94 |
| DeepSVDD | 0.8053 | 0.5840 | 6.62 | 0.13 | 0.23 |
| GOAD | 0.7210 | 0.5225 | 9.55 | 0.02 | 0.57 |
| NeuTralAD | 0.8391 | 0.6262 | 5.62 | 0.23 | 0.23 |
| ICL | 0.8492 | 0.6551 | 5.26 | 0.23 | 0.11 |
| MCM | 0.8166 | 0.5988 | 6.40 | 0.13 | 0.30 |
| NF-SLT | **0.8691** | **0.6749** | **4.02** | **0.30** | **0.09** |

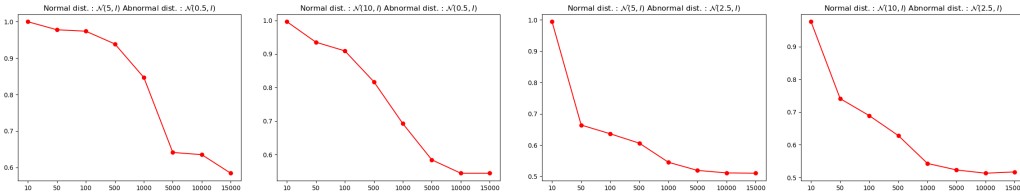

Figure 1: Performance of NF-SLT across different dimensions. The y-axis represents AUROC, and the x-axis indicates the dimensionality of the data. The titles of each subfigure specify whether the data follows a normal or abnormal distribution.

difference in AUROC score according to dimension was recorded in Figure 1. The dimension was set to [10, 50, 100, 500, 1000, 5000, 10000, 15000].

In Figure 1, in the 1st and 2nd subfigures, performance degradation occurs sharply from 1000 to 5000. The 3rd and 4th subfigure begin to degrade in performance at a dimension of 50, smaller than the first two figures, and continue to degrade as the dimension increases. Histograms of the log-likelihood of the normal and abnormal data for each experimental setting are recorded in Appendix F.

In the followings, we will review some distributional properties of the Euclidean norm and demonstrate why this characteristic makes anomaly detection using normalizing flows particularly fails as dimensionality increases.

### 5.1 EUCLIDEAN NORM GETS MORE CONCENTRATED AS DIMENSION INCREASES.

To clarify the motivation for examining the distributional characteristics of the Euclidean norm, we provide a brief review of the normalizing flow model. The normalizing flow model $f$ is trained on normal data $X$ such that $f(X)$ follows a standard Gaussian distribution, i.e., $f(X) \sim N(0, I)$. The training objective is to maximize the log-likelihood, $\log P_{N(0,I)}(\Theta \mid f(X)) \propto -\|f(X)\|_2^2$, where $\Theta$ represents the parameters of the normalizing flow and $\|\cdot\|_2$ denotes the Euclidean norm. If $Z \sim N(0, I_d)$, then $\|Z\|_2^2$ concentrates near $d$. As a result, the test statistic $\|f(X)\|_2^2$ is close to $d$ if $X$ is normal data. If the normalizing flow is well-trained, the transformed normal data $f(X)$ will be concentrated on a sphere of radius $\sqrt{d}$ (where $d$ is the dimensionality of $X$) due to the tail bound properties of sum of independent normal distributions.

The concentration behavior of the Euclidean norm in high dimensions plays a key role in understanding the performance of normalizing flows in distinguishing between normal and anomalous data. We now turn to the formal mathematical reasoning that supports this observation.

**Proposition 1.** *If $Z \sim N(0, I_d)$, then for all $0 < t < d$ :*

$$\Pr\left(\left|||Z||_2^2 - d\right| \geq t\right) \leq 2e^{-\frac{t^2}{8d}} \tag{6}$$

This result indicates that the Euclidean norm of a standard normal distribution tends to concentrate around the dimension d as the dimensionality increases.

*Proof.* Take a random variable $Z \sim N(0, I_d)$ in $\mathbb{R}^d$. Then for each $Z_i \sim N(0, 1)$, $\mathbb{E}\left[e^{\lambda(Z_i^2 - 1)}\right] = \frac{e^{-\lambda}}{\sqrt{1-2\lambda}} \leq e^{4\lambda^2/2}$ for all $|\lambda| < 1/4$. Thus, $||Z||_2^2$ is sub-exponential with parameters $(2\sqrt{d}, 4)$ and by the properties of sub-exponential random variables, we obtain the concentration bound : $\Pr\left(\left|||Z||_2^2 - d\right| \geq t\right) \leq 2e^{-\frac{t^2}{8d}}$ for $0 < t < d$.

$\square$

**Definition 2.** *A random variable X is isotropic if $\mathbb{E}(X) = 0$ and $\mathbb{E}(XX^T) = I_d$*

**Definition 3.** *A random variable X is log-concave if its density function $f(x) \propto e^{-H(x)}$ where H is a convex function.*

**Theorem 1** (Klartag (2007), Fleury et al. (2007),Guédon (2014))**.** *X is a log-concave isotropic random variable in $\mathbb{R}^d$.*

*If $\exists \epsilon_d(\epsilon_d \to 0)$ such that $\Pr\left(\left|\frac{||X||_2}{\sqrt{d}} - 1\right| \geq \epsilon_d\right) \leq \epsilon_d$, then $\lim_{d\to\infty} \frac{\text{Var}||X||_2}{d} = 0$*

**Proposition 2.** *If $Z \sim N(0, I_d)$,*

$$\lim_{d\to\infty} \frac{\text{Var}||Z||_2}{d} = 0 \tag{7}$$

*Proof.* From the proposition 1, take $t = dt$. Then $\Pr\left(\left|||Z||_2^2 - d\right| \geq dt\right) \leq 2e^{-\frac{dt^2}{8}}$ for $0 < t < 1$.

Since there exists $\epsilon_d$ such that $\max\{t, 2e^{-\frac{dt^2}{8}}\} < \epsilon_d$ and $\epsilon_d \to 0$ and $\Pr\left(\left|\frac{||Z||_2}{\sqrt{d}} - 1\right| \geq t\right) = \Pr\left(\left|\frac{||Z||_2^2}{d} - 1\right| \geq t\right)$, there exists $\epsilon_d(\epsilon_d \to 0)$ such that $\Pr\left(\left|\frac{||Z||_2}{\sqrt{d}} - 1\right| \geq \epsilon_d\right) \leq \epsilon_d$. By theorem 1, $\lim_{d\to\infty} \frac{\text{Var}||Z||_2}{d} = 0$

$\square$

Therefore, based on the propositions above, as dimensionality increases, the Euclidean norm of a normal random variable tends to concentrate more quickly relative to the increase in dimensionality. This effect exacerbates the issue described in (Zhang et al., 2021): even if a model approximates a perfectly normal distribution (meaning the data is not anomalous), a slight misestimation in density can lead the model to assign a higher likelihood to data points from an abnormal distribution.

## 5.2 EUCLIDEAN NORM IS ALMOST IDENTICAL IN HIGH-DIMENSIONAL SPACE

The Euclidean norm becomes an ineffective measure for statistical testing as dimensionality increases because data points from different distributions tend to have nearly identical norms. As dimensionality rises, the Euclidean norms of data points concentrate around similar values, even when the points are generated from distinct distributions. This makes it difficult to distinguish between normal and abnormal data based solely on the Euclidean norm in high-dimensional spaces.

**Theorem 2** (Berry-Essen Type Inequality for convex Body)**.** *If $X$ is uniformly distributed in a convex body $K \subset \mathbb{R}^d$, then*

$$\mathbb{E}\left(||X||_2 - \sqrt{d}\right)^2 \leq C^2$$

**Theorem 3** (Guédon & Milman (2011))**.** *For a log-concave and isotropic random variable $X$ in $\mathbb{R}^d$, there exists a constant $C$ such that for any $t > 0$,*

$$\Pr\left(||X||_2 - \sqrt{d} \geq t\sqrt{d}\right) \leq Ce^{-c\sqrt{d}\min\{t^3, t\}}$$

**Conjecture 1** (Thin-Shell Conjecture). *For a log-concave and isotropic random variable $X$ in $\mathbb{R}^d$, there exists a constant $C$ such that for any $t > 0$,*

$$\Pr\left(\left|\|X\|_2 - \sqrt{d}\right| \geq t\sqrt{d}\right) \leq 2e^{-Ct\sqrt{d}}$$

Although the Thin-Shell conjecture has not yet been proven, there have been several breakthroughs by the works including Eldan (2013) and Chen (2021). As the Thin-Shell Conjecture and the results of Guédon & Milman (2011) show, all the log-concave and isotropic random variables have their Euclidean norm near $\sqrt{d}$.

**Theorem 4** (Anttila et al. (2003)). *$X$ is log-concave isotropic random variable in $\mathbb{R}^d$. If there exists $\epsilon_d \to 0$ as $d \to \infty$ such that $\Pr\left(\left|\frac{\|X\|_2}{\sqrt{d}} - 1\right| \geq \epsilon_d\right) \leq \epsilon_d$, then there exists $\theta \in S^{d-1}$*

$$\sup_{t>0}\left|\Pr\left(\sum_{i=1}^d \theta_i X_i \leq t\right) - \frac{1}{\sqrt{2\pi}}\int_{-\infty}^t e^{-v^2/2}dv\right| \leq \eta_d$$

*, where $\eta_d \to 0$*

This theorem by Anttila et al. (2003) demonstrates that if the Euclidean norm of a random variable in $\mathbb{R}^d$ concentrates near $\sqrt{d}$, there exists a linear functional of X that closely approximates a normal distribution. Klartag (2007) extended this result, showing that almost every linear functional of $X$ becomes approximately normally distributed as $d \to \infty$. These results imply that in high-dimensional spaces, the concentration of the Euclidean norm is nearly identical across distributions, which reduces the effectiveness of hypothesis tests based on the Euclidean norm in distinguishing between distributions. In fact, as shown in the experimental results in Appendix F, the likelihood histograms reveal that although the normal and anomaly data are clearly derived from different distributions, the distributions of their likelihoods overlap as the dimensionality increases.

## 6 LIMITATION

We demonstrate that counterintuitive phenomena rarely occur in tabular anomaly detection by showing that NF-SLT outperforms most shallow and deep models. However, it still fails to beat the performance of $k$-NN, a strong baseline. In addition, there is a limitation in that the experiments were conducted only using relatively simple models, NICE and RealNVP, rather than advanced normalizing flow models. However, our study focuses on that counterintuitive phenomena rarely occur when performing simple likelihood tests in tabular domains and provides empirical and theoretical explanations for why "**counter of counterintuitive**" phenomena. Therefore, in addition to comparing the anomaly detection performance with advanced normalizing flow models, future work could be to propose a flow architecture that can outperform $k$-NN in the tabular domain compared to the current flow model architecture suitable for the image domain like RealNVP's masking strategy.

## 7 CONCLUSION

In this paper, we investigated whether the counterintuitive phenomenon observed in anomaly detection for image data also occurs in the tabular domain. Through both theoretical analysis and extensive experiments, we showed that this phenomenon is rare in tabular data when using simple likelihood tests with normalizing flow. Our results demonstrate that normalizing flow-based methods are highly effective for tabular anomaly detection, outperforming traditional models without encountering the issues seen in the image domain.

We also addressed the problem of biased hyperparameter selection in previous studies, proposing a fair and consistent evaluation framework. Based on our findings, future research should focus on developing normalizing flow architectures specifically designed for tabular data and ensuring fair hyperparameter selection in unsupervised settings. Our work provides a strong foundation for advancing anomaly detection techniques in structured data domains.

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
