# OpenReview forum: "Challenging the Counterintuitive: Revisiting Simple Likelihood Tests with Normalizing Flows for Tabular Data Anomaly Detection"
_ICLR.cc/2025/Conference — ICLR 2025 Conference Withdrawn Submission_

### Official Review · Reviewer_ZzYs · 2024-10-21

**Soundness:** 2
**Presentation:** 3
**Contribution:** 2
**Rating:** 3
**Confidence:** 4

**Summary:**

This paper combines existing normalizing flows based methods with a simple likelihood test for anomaly detection in tabular domain. The authors also redefine the counterintuitive phenomenon, which deteriorates the performance of normalizing flows based methods in image domain,  and demonstrate, both theoretically and empirically, why this method succeeds in the tabular domain. To avoid biased and impractical hyper-parameter selection, this paper leverages grid search and adopts the hyperparameter combination with the highest performance for each comparison methods. The experimental results are encouraging compared to baselines on all 47 tabular datasets presented in ADBench. This paper also discusses the impact of the Euclidean norm on the method as the data dimension increases.

**Strengths:**

This paper can provide insights into the advancement of normalizing flows methods for anomaly detection, the discussion on the impact of data dimensionality to the simple likelihood tests in tabular data is meaningful, the experimental results are unbiased and promising.

**Weaknesses:**

1. The article only discusses the impact of the data dimension on the success of the simple likelihood testing using normalizing flows in the tabular domain. However, the difference between tabular data and image data is not just that the dimensionality is lower. For example, the features of image data are homogeneous, highly correlated, while the features of tabular data are heterogeneous, some features are totally irrelevant. This paper lacks the analysis of the impact of these other differences on the counterintuitive phenomenon. When the feature dimensions are the same, what is the difference between tabular data and image data in terms of counterintuitive phenomenon?

2. The method in this paper is just an application of existing normalizing flow models rather other presenting a new method, which limits the novelty of the paper. Besides, as mentioned in the paper, the experiments are conducted only using relatively simple models, NICE and RealNVP, rather than advanced normalizing flow models, will employing an advanced model further improve the performance?

3. One contribution in this paper is that it conducts fair hyperparameter selection, however, the paper just leverage a simple grid search for each comparison method, exploring the performance and difference of other hyperparameter selection methods could enrich the analysis.

**Questions:**

See Above.

---

### Official Review · Reviewer_ZVSY · 2024-10-26

**Soundness:** 2
**Presentation:** 3
**Contribution:** 2
**Rating:** 3
**Confidence:** 5

**Summary:**

This paper aims to explain why AD using generative models works tabular data but fails on images. It first performs an extensive and careful comparison between methods showing that differently from some previous reports, kNN performs the best on tabular AD and inverse flows second. It then explain the apparent divergence from the image AD results using euclidean norm concentration arguments.

**Strengths:**

The reviewer believes the main contribution in this paper is the careful and fair comparison of tabular AD approaches, The field has been plagued by unfair evaluation making the true SoTA unclear. Showing that simple kNN remains the best method is important. Note that past papers also showed this e.g. [1] but as many methods appeared since that claimed to be better than kNN, it is useful to have an up-to-date evaluation. The reviewer would have preferred this to be the focus of the paper.

[1] Statistical Analysis of Nearest Neighbor Methods for Anomaly Detection , Gu et al,, NeuIPS'19

**Weaknesses:**

The reviewer believes the premise of this paper is unsound. The claim is that images suffer from the "counterintuitive phenomenon" due to high dimension, while tabular data do not due to lower dimension. However, the story is more nuanced. The generative models here were estimated on image pixels. The story however would be completely different if the images were first pre-processed by a pretrained deep feature extractor (as is standard in image AD). In that case, kNN based methods achieve SoTA performance (e.g., PANDA or PatchCore) and inverse flow methods on the deep features perform comparably. This is despite the fact that the feature dimension is very large.

The reviewer believes that current empirical evidence points to a different direction from the one proposed here. The main issue with kNN / likelihood methods is not the dimensionality of the data, but rather the quality of the representation. Tabular datasets typically have human engineered features which are excellent representations. Even simple L2 distance between raw tabular features is often related to semantic difference. Pixels are not semantic image representations. L2 distance between image pixels is not well correlated with semantic distance. Deep pretrained image features are again excellent semantic representations. This explains why kNN (and therefore also likelihood methods which compute the PDF is the representation space) behave the way they do . The result is therefore not particularly surprising, and the provided explanation is probably not the most salient one.

**Questions:**

The reviewer provided an alternative explanation for the phenomenon. The rebuttal should challenge this explanation or convince in some other way why the story is dimension and not the quality of representation.

---

### Official Review · Reviewer_UTYB · 2024-11-02

**Soundness:** 2
**Presentation:** 2
**Contribution:** 2
**Rating:** 3
**Confidence:** 3

**Summary:**

This paper explores the use of normalizing flow for tabular anomaly detection, and the experiments appear to demonstrate the effectiveness of the proposed method.

**Strengths:**

1、The paper offers a theoretical analysis of the proposed NF-SLT.

2、Extensive experiments are conducted to demonstrate the effectiveness of NF-SLT.

**Weaknesses:**

1、The authors claim that normalizing flow fails to perform anomaly detection for images. However, many studies in unsupervised anomaly detection achieve state-of-the-art results in detecting visual anomalies, such as Fastflow [1] and Cflow-ad [2].

2、The performance comparisons are limited. NF-SLT uses a relatively strong normalizing flow as the backbone. However, these baselines are lighter and are not SOTAs in this field. The comparisons are not convincing. More importantly, there are SOTA normalizing flow-based anomaly detection methods, such as GANF [3] and MTGFlow [4], which should be included to make a meaningful comparison.

3、Could you provide visualizations to demonstrate that the log-likelihood can be regarded as an anomaly indicator? Relying solely on these quantitative results fails to present an intuitive advantage over NF-SLT.

[1] Yu1, J., Zheng, Y., Wang, X., Li, W., Wu, Y., Zhao, R., & Wu, L. (2021). FastFlow: Unsupervised Anomaly Detection and Localization via 2D Normalizing Flows. ArXiv, abs/2111.07677.

[2] Gudovskiy, D.A., Ishizaka, S., & Kozuka, K. (2021). CFLOW-AD: Real-Time Unsupervised Anomaly Detection with Localization via Conditional Normalizing Flows. 2022 IEEE/CVF Winter Conference on Applications of Computer Vision (WACV), 1819-1828.

[3] Dai, E., & Chen, J. (2022). Graph-Augmented Normalizing Flows for Anomaly Detection of Multiple Time Series. ArXiv, abs/2202.07857.

[4] Zhou, Q., Chen, J., Liu, H., He, S., & Meng, W. (2022). Detecting Multivariate Time Series Anomalies with Zero Known Label. AAAI Conference on Artificial Intelligence.

**Questions:**

See weaknesses. I will consider improving my rating if the authors could address my concerns.

---

### Official Review · Reviewer_RCS5 · 2024-11-03

**Soundness:** 2
**Presentation:** 2
**Contribution:** 2
**Rating:** 3
**Confidence:** 4

**Summary:**

This paper solves tabular anomaly detection with normalizing flow model and formalizes the counterintuitive phenomenon.

**Strengths:**

Introduces the normalizing flow model for tabular anomaly detection.

**Weaknesses:**

Please refer to questions below

**Questions:**

• The method in the paper does not elaborate on the implementation details, which is not reader-friendly.
• In Section 5, only show the results on synthetic datasets, such as cardio and cardiotocography, cover and donors, ionosphere and letter, and so on, these datasets
in the same dimension, why not use real datasets for analysis like [1]?
[1] Why Normalizing Flows Fail to Detect Out-of-Distribution Data. NeurIPS 2020.
• Compared with the results reported by MCM[2], the effect of using Fair hyperparameter is quite different. If fine-tuning is used, why not use the best results?
[2] MCM: Masked Cell Modeling for Anomaly Detection in Tabular Data. ICLR 2024.
• I'd also like to see NPT-AD [3] performance results, as these are not presented in the paper.
[3] Beyond Individual Input for Deep Anomaly Detection on Tabular Data. ICML 2024.
• What is the time cost and computational cost of different models？

---

### Official Review · Reviewer_SokV · 2024-11-05

**Soundness:** 1
**Presentation:** 2
**Contribution:** 1
**Rating:** 3
**Confidence:** 4

**Summary:**

Deep generative models like normalizing flows have shown counterintuitive behavior when detecting anomalies in image data (Nalisnick et al., 2018). The paper considers using normalizing flows to detect anomalies but in the tabular data domain. The paper doesn’t find the counterintuitive observation re-occurs. The authors claim normalizing flows are effective methods for real-world tabular data anomaly detection and demonstrate the phenomenon empirically through one specific model. The authors borrow the properties of Euclidean norms in high dimensional space to explain why normalizing flows fail in high dimensional spaces.

**Strengths:**

- The paper has done large-scale experiments on ADBench datasets. A lot of methods have been evaluated in the same setting.
- The limitation section covers some of the concerning aspects.

**Weaknesses:**

My main concern comes from the reasoning of the claim
- There are ambiguities in definition 1. According to it, does OCSVM have counterintuitive performance as it is rated to be the worst among compared methods?
- The empirical evidence is weak. The paper should discuss other normalizing flows rather than NF-SLT in Table 1.
- The theoretical analysis only focuses on high-dimension curse and is not specific to flow methods. Moreover, it doesn’t clearly state why flow methods succeed in tabular data anomaly detection.

**Questions:**

Here are some questions / minor concerns:
- How about the original definition – OOD has higher likelihood than ID? Why don’t you use that definition?
- Does fef 1 consider model complexity?
- Overclaimed? L226: “To the best of our knowledge, this is the first time we have run an experiment with all the tabular data proposed in ADBench…” A lot of papers have grounded their experiments on the whole ADBench.
- What makes the difference between subplots 3, 4 and subplots 1, 2 in Fig 1?
- L159: “...higher likelihoods to OOD data…” should be “lower”

---

### Note · Authors · 2024-11-16

I have read and agree with the venue's withdrawal policy on behalf of myself and my co-authors.